# Predictive factors for readmission due to neonatal hyperbilirubinemia: A retrospective case-control study

Yueju Cai[1] *, Xiaolan Li[2], Ping Wang[1], Yanyan Song[2]

**1** Department of Neonatology, Guangzhou Wowen and Children's Medical Center, Guangzhou Medical University, Guangzhou, China, **2** Department of Child Health, Guangzhou Women and Children's Medical Center, Guangzhou Medical University, Guangzhou, China

☯ These authors contributed equally to this work and should be considered as co-first authors.
\* caiyueju0323@163.com

## Abstract

### Objective

Hyperbilirubinemia is a common cause of hospital readmission among neonates, but the factors contributing to post-discharge readmission remain unclear. Our study aimed to identify predictive factors associated with readmission for neonatal hyperbilirubinemia.

### Methods

This retrospective case-control study included 421 neonates born at ≥35 weeks of gestation with hyperbilirubinemia between January and December 2021. The neonates were divided into a readmission group and a control group, and logistic regression was used to identify predictive factors associated with readmission.

### Results

Among the 421 neonates studied, 32 (7.6%) were readmitted. Logistic regression analysis identified preterm birth (<37 weeks), ABO hemolysis, Glucose-6-Phosphate Dehydrogenase (G6PD) deficiency, and Total Serum Bilirubin (TSB) level at discharge as significant predictive factors for readmission due to hyperbilirubinemia in newborns. Additionally, a decrease in birth weight was significantly linked to an increased risk of readmission (OR = 0.998, P = 0.013), although the effect size was relatively small.

### Conclusions

Prolonging hospitalization and implementing robust post-discharge monitoring may be essential for neonates with prematurity, ABO hemolysis, G6PD deficiency, or elevated TSB levels at discharge.

## Introduction

Hyperbilirubinemia is a common cause of readmission in neonates, with approximately 60% of full-term and 80% of preterm infants presenting with jaundice within the first few days after birth

**Data availability statement:** All relevant data are within the manuscript and its Supporting Information files.

**Funding:** This study was supported by the Guangzhou Health Science and Technology Project (Grant No. 20241A011025), the Guangdong Medical Science and Technology Research Foundation (Grant No. A2023164), and the Liuzhou Science and Technology Planning Project (Grant No. 2024SB0104A002). The recipient of the funding awards listed above was Cai Yueju. The funders had no role in study design, data collection and analysis, decision to publish, or preparation of the manuscript.

**Competing interests:** The authors have declared that no competing interests exist.

[1]. While most cases are effectively managed without significant complications, severe hyperbilirubinemia can result in permanent neurological disorders such as acute bilirubin encephalopathy and Kernicterus, resulting in limited treatment options [2]. Early discontinuation of phototherapy may increase the risk of rebound hyperbilirubinemia and subsequent readmission, while prolonged use of phototherapy can result in increased hospital costs and maternal-infant separation [3]. Additionally, studies have indicated a potential correlation between phototherapy and melanocyte nevus as well as tumors in infants [4,5]. Previous studies have reported the risk factors of rebound hyperbilirubinemia during hospitalization and predicting its occurrence [6–8]. Nevertheless, research concerning the risk factors related to readmission after discharge remains inadequate. The study by Xiao et al [9]. demonstrated that low gestational age, younger age at initial admission, and G6PD deficiency were independent risk factors for the readmission of neonates with hyperbilirubinemia. Hence, this study is designed to explore the predictive factors for readmission of neonates with hyperbilirubinemia after discharge.

## Participants and methods

### Participants

A retrospective case-control study was conducted on neonates with hyperbilirubinemia who were admitted to the Women and Children's Medical Center of Guangzhou Medical University from January to December 2021. The inclusion criteria encompassed newborns with a gestational age of no less than 35 weeks and a confirmed diagnosis of hyperbilirubinemia (refer to Fig 1).

### Inclusion criteria

The inclusion criteria are as follows: 1) a gestational age of 35 weeks or greater; 2) predominance of indirect bilirubin; 3) the necessity for phototherapy or exchange transfusion; and 4) first hospitalization due to jaundice.

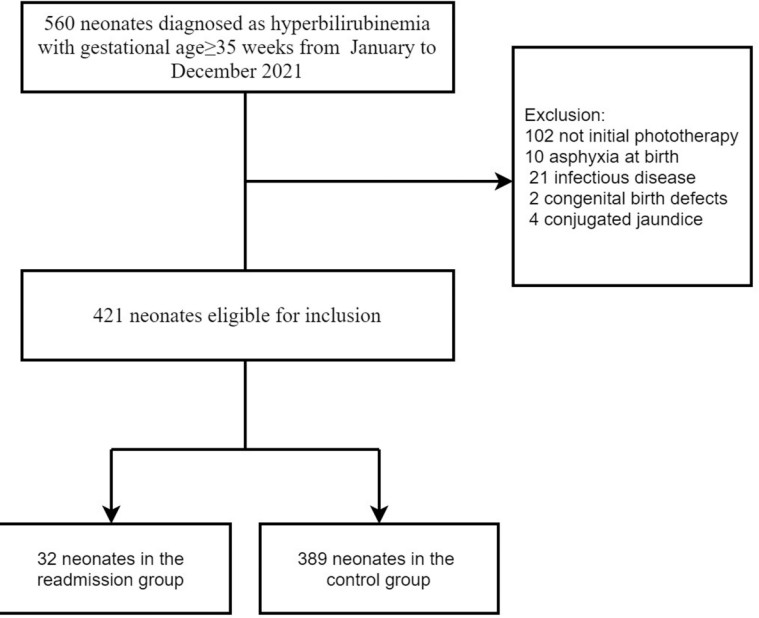

**Fig 1. Flow of participation for the neonates with hyperbilirubinemia.**

## Exclusion criteria

The exclusion criteria for this study include congenital malformations, metabolic disorders, conjugated hyperbilirubinemia, and conditions such as hypoxia or infectious diseases. Additionally, infants who have received phototherapy for neonatal hyperbilirubinemia at other institutions are also excluded.

## Methods

Data collection was completed between October and December 2021, and we also had access to information that could identify individual participants during or after data collection. Perinatal data were collected from for all infants included in the study, who were subsequently categorized into two groups based on readmission status: a readmission group and a control group. A comparative analysis was conducted between the two groups on various factors, including basic demographic characteristics (sex, gestational age, birth weight, mode of delivery, maternal complications, etc.), time at first admission, jaundice within 24 hours of birth, total serum bilirubin (TSB) at first admission, duration of phototherapy, TSB at first discharge, blood group incompatibility (both ABO and Rh), immunoglobulin administration, glucose-6 -phosphate dehydrogenase (G6PD) status, cephalohematoma occurrence, polycythemia incidence, exchange transfusion necessity, age at first discharge and length of initial hospitalization.

Phototherapy and exchange transfusion are performed in accordance with established guidelines [10]. Transcutaneous bilirubin (TcB) is utilized for the screening of neonatal jaundice. However, in instances where TcB reaches or exceeds the hour-specific threshold, or when there is a risk of hemolysis and sepsis, or if jaundice is progressing,TSB levels are measured. If the infant's TSB level reaches or exceeds the escalation-of-care threshold, defined as 2 mg/dL below the exchange transfusion threshold, TSB measurements are performed at intervals of no less than two hours until the end of the escalation-of-care period. Discontinuing phototherapy is an option when the TSB has decreased by at least 2 mg/dL below the hour-specific threshold at the initiation of phototherapy. Infants who received phototherapy prior to 48 hours of age, exhibited a positive direct antiglobulin test (DAT), or had known or suspected hemolytic disease should have their TSB levels measured 6 to 12 hours following the discontinuation of phototherapy, with a repeat bilirubin assessment conducted on the day subsequent to phototherapy cessation. For all other infants, bilirubin levels are assessed on the day after phototherapy discontinuation.If the TSB levels of the infants remain below the hour-specific threshold for a duration of 24 hours following the discontinuation of phototherapy, they will be eligible for discharge.

The current study received approval from the Ethics Committee of Women and Children Medical Center of Guangzhou Medical University (No. 178A01) in accordance with ethical guidelines. For this retrospective study, informed consent was waived as sanctioned.

## Statistical analysis

The statistical analysis was conducted using IBM Corp.'s SPSS software, version 25.0, in Armonk, NY. The data were presented either as mean and standard deviation or median and interquartile range (IQR), depending on their distribution. Frequency measures were expressed as numbers (n) and percentages (%). A comparison was made between the readmission group and control groups based on demographic characteristics and outcomes. The independent sample t-test was utilized for normally distributed variables, while the Mann–Whitney U-test was employed for non-normally distributed variables to assess group differences. The Chi-square test was used to analyze categorical variables. Logistic regression

analysis was performed to determine the risks of readmission. A significance level of p < .05 was considered statistically significant.

In this study, we initially attempted to categorize birth weight into three groups: low birth weight (<2500g), normal weight (2500–4000g), and high birth weight (>4000g) for clinical interpretability. However, due to the extremely small sample sizes in the low birth weight and high birth weight groups (4 cases each), we decided to include birth weight as a continuous variable in the regression model to ensure robust and reliable statistical results.

Gestational age was initially included in the regression model in both continuous (in weeks) and categorical (preterm < 37 weeks vs. term ≥ 37 weeks) forms to explore its impact on the risk of readmission. However, the use of both forms in the model raised concerns about multicollinearity and model stability. Given that clinicians are more focused on preterm infants as an independent high-risk group rather than subtle weekly changes in gestational age, we decided to retain only the categorical form. After this adjustment, multicollinearity tests were re-conducted for other independent variables to ensure the robustness of the final model.

## Results

A total of 421 infants diagnosed with hyperbilirubinemia between $35^{+0}$ and $41^{+6}$ weeks of gestational age were included in the study. Among these, 208 infants (49.4%) were male and 213 (50.6%) were female. Within this cohort, 18 infants (4.3%) were preterm, with a median gestational age of 36.5 weeks (interquartile range: 35.2 to 36.6) and a birth weight of 2,855 grams (interquartile range: 2,290 to 3,280). Of the total population studied, 32 newborns (7.6%) required readmission, constituting the readmission group; the remaining 389 neonates comprised the control group. The median interval for readmission was found to be 5.0 days (interquartile range: 3.0 to 6.0). Notably, both gestational age and birth weight were significantly lower in the readmission group compared to those in the control group (p < .05). Furthermore, the readmission group exhibited a higher proportion of premature infants relative to the control group (p < .05) (Table 1).

Compared to the control group, the readmission group exhibited an earlier time of first admission, with median hours of 48.5 (interquartile range: 26.0 to 61.5) and 53.0 (interquartile range: 39.5 to 87.0), respectively (p = 0.03). Jaundice onset occurred within 24 hours of birth in 173 infants (41.1%), with a significantly higher proportion in the readmission group (p = 0.045). The readmission group also demonstrated a greater incidence of ABO hemolysis compared to controls: 14 out of 32 infants (43.8%) versus 94 out of 389 infants (24.2%), respectively (p = 0.015), as well as a higher incidence of glucose-6-phosphate dehydrogenase (G6PD) deficiency:12 out of 32 infants (37.5%) versus 83 out of 389 infants(21.3%), respectively(p = 0.036). Additionally, infants in the readmission group required a longer duration of phototherapy: 30.0 hours (interquartile range: 24.0 to 48.0) versus 24.0 hours (interquartile range:18.0 to 36.0),respectively(p = 0.013). Upon initial discharge, TSB levels were significantly elevated in the readmission group (p = 0.000). No significant differences were observed between groups regarding TSB at first admission, immunoglobulin administration, cephalohematoma occurrence, polycythemia incidence, age at first discharge or length of initial hospitalization. (Table 2).

After excluding the continuous form of gestational age from the model, multicollinearity tests were re-conducted for the remaining independent variables. The final model showed no evidence of significant multicollinearity, ensuring robust and reliable results. As presented in Table 3, the logistic regression analysis identified preterm birth (< 37 weeks), ABO hemolysis,

**Table 1. Comparison of demographic features.**

| Variables | All patients (n = 421) | Readmitted group (n = 32) | Control group (n = 389) | P values |
|---|---|---|---|---|
| Male Sex, n (%) | 208 (49.4) | 19 (59.4) | 189 (48.6) | 0.241 |
| Gestational age (w), mean (SD) | 39.0 (1.2) | 38.1 (1.5) | 39.0 (1.1) | 0.002 |
| Preterm, n (%) | 18 (4.3) | 7 (21.9) | 11 (2.8) | 0.000 |
| Multiple pregnancy, n (%) | 18 (4.3) | 2 (6.3) | 16 (4.1) | 0.404 |
| Birth weight (g), mean (SD) | 3186 (356) | 2981 (295) | 3202 (356) | 0.001 |
| Caesarean Delivery, n (%) | 103 (24.5) | 6 (18.8) | 97 (24.9) | 0.434 |
| Stained amniotic fluid, n (%) | 76 (18.1) | 6 (18.8) | 70 (18.0) | 0.915 |
| Exclusive Breastfeeding, n (%) | 155 (36.8) | 10 (31.3) | 145 (37.3) | 0.497 |
| Maternal complications, n (%) | | | | |
| PROM | 114 (27.1) | 8 (25.0) | 106 (27.2) | 0.783 |
| GDM | 85 (20.2) | 6 (18.8) | 79 (20.3) | 0.833 |
| ICP | 6 (1.4) | 1 (3.1) | 5 (1.3) | 0.380 |
| Length of first hospitalization (IQR), days | 5.0 (4.0-6.0) | 5.0 (4.0-7.0) | 5.0 (4.0-6.0) | 0.311 |

PROM: Premature Rupture of Membranes; GDM: Gestational Diabetes Mellitus; ICP: Intrahepatic Cholestasis of Pregnancy.

**Table 2. Comparison of examination test and treatment.**

| Variables | All patients (n = 421) | Readmitted group (n = 32) | Control group (n = 389) | P values |
|---|---|---|---|---|
| Time at first admission (IQR), hours | 53.0 (38.5-84.0) | 48.5 (26.0-61.5) | 53.0 (39.5-87.0) | 0.03 |
| Jaundice within 24 hours of birth, n (%) | 173 (41.1) | 19 (59.4) | 154 (39.6) | 0.029 |
| TSB at first admission (IQR), umol/L | 231.0 (192.8-296.3) | 239.5 (207.0-308.6) | 230.5 (192.5-296.3) | 0.384 |
| Duration of phototherapy (IQR), hours | 24.0 (18.0-38.0) | 30.0 (24.0-48.0) | 24.0 (18.0-36.0) | 0.013 |
| TSB at first discharge (IQR), umol/L | 187.6 (166.5-206.7) | 203.8 (192.6-232.6) | 185.9 (164.2-204.3) | 0.000 |
| ABO hemolysis, n (%) | 109 (25.9) | 15 (46.9) | 94 (24.2) | 0.005 |
| Immunoglobulin administration, n (%) | 15 (3.6) | 2 (6.3) | 13 (3.3) | 0.318 |
| G6PD deficiency, n (%) | 95 (22.6) | 12 (37.5) | 83 (21.3) | 0.036 |
| Cephalohematoma, n (%) | 21 (5.0) | 1 (3.1) | 20 (5.1) | 0.514 |
| Polycythemia, n (%) | 17 (4.0) | 2 (6.3) | 15 (3.9) | 0.376 |
| Age of first discharge (IQR), days | 7.0 (6.0-9.0) | 8.0 (6.0-10.0) | 7.0 (6.0-9.0) | 0.175 |

G6PD deficiency, and elevated TSB levels at discharge as independent risk factors significantly associated with readmission for neonatal hyperbilirubinemia. Additionally, a decrease in birth weight was significantly linked to an increased risk of readmission (OR = 0.998, P = 0.013), although the effect size was relatively small.

## Discussion

In this study, we observed a readmission rate of 7.6% for neonatal hyperbilirubinemia following discharge. Logistic regression analysis identified preterm birth(< 37 weeks), ABO hemolysis, G6PD deficiency, and elevated TSB levels at first discharge as significant independent risk factors for readmission due to neonatal hyperbilirubinemia. Additionally, a decrease in birth weight was significantly associated with an increased risk of readmission (OR = 0.998, P = 0.013), although the effect size was relatively small.

In the study conducted by Belide et al., preterm birth (gestational age < 37 weeks) and lower birth weight were identified as significant risk factors for rebound hyperbilirubinemia

**Table 3. Logistic regression model to predict readmission risk factors.**

| Variables | B | SE | P values | OR |
|---|---|---|---|---|
| Preterm birth (<37) | 2.256 | 0.645 | <0.001 | 9.541 (2.693-33.796) |
| Birth weight | -0.002 | 0.001 | 0.013 | 0.998 (0.997-1.000) |
| Onset age at phototherapy | -0.020 | 0.011 | 0.077 | 0.981 (0.959-1.002) |
| Jaundice within 24 hours of birth | -0.149 | 0.548 | 0.786 | 0.861 (0.294-2.523) |
| Duration of phototherapy | -0.001 | 0.010 | 0.931 | 0.999 (0.980-1.019) |
| TSB level at discharge | 0.032 | 0.008 | <0.001 | 1.032 (1.016-1.049) |
| ABO hemolysis | 1.103 | 0.498 | 0.027 | 3.013 (1.135-7.998) |
| G6PD deficiency | 1.188 | 0.484 | 0.014 | 3.281 (1.270-8.472) |

[11]. This study found that preterm birth is one of the most significant predictors of readmission for neonatal hyperbilirubinemia. Compared to term infants, the risk of readmission is significantly higher in preterm infants (OR = 9.541). This may be attributed to the insufficient bilirubin metabolism capacity in preterm infants, such as the immaturity of the UDPGT enzyme and reduced hepatic clearance. Therefore, stricter monitoring and management of bilirubin levels in preterm infants are essential in clinical practice. Regarding birth weight, we initially considered categorizing it to enhance clinical interpretability. However, the low sample sizes in the low birth weight group (<2500g) and high birth weight group (>4000g) rendered the statistical analysis of categorized data insufficiently meaningful. As a result, we opted to treat birth weight as a continuous variable, which provided more robust and reliable results. Our analysis revealed a significant association between decreased birth weight and an increased risk of readmission (OR = 0.998, P = 0.013), though the effect size is small. The increased risk of readmission in low birth weight infants may be attributed to their overall lower developmental levels.

Although jaundice occurring within the first 24 hours of life may not have an identifiable cause, when a cause is determined, it is most likely attributable to a hemolytic process. Our study found that a greater proportion of infants in the readmission group experienced jaundice within this timeframe, potentially linked to lower gestational age or factors such as ABO hemolysis. Furthermore, several studies have established a correlation between rebound hyperbilirubinemia in neonates and hemolytic diseases, particularly ABO hemolytic, which may prolong hospitalization and extend the duration of phototherapy [12,13]. In our study, ABO hemolysis was identified as a risk factor for readmission, which is consistent with the findings of Xu et al. [13] who reported a readmission rate of 12.4% (36/291) in neonates with ABO hemolytic disease. Consequently, TSB or TcB levels should be promptly measured in infants exhibiting jaundice within the first 24 hours after birth, particularly in those with a gestational age of less than 37 weeks or suspected ABO hemolysis.

G6PD deficiency is now widely acknowledged as a significant contributor to severe hyperbilirubinemia [14]. Research indicates that approximately 13% of African American males and 4% of African American females are affected by G6PD deficiency [15]. Infants with G6PD deficiency face an elevated risk of readmission and retreatment for hyperbilirubinemia [16]. Our research demonstrates that G6PD deficiency independently elevates the likelihood of readmission due to hyperbilirubinemia, consistent with findings from previous studies [17]. In regions such as Guangdong Province in China, where G6PD deficiency is prevalent, practices like Chinese herbal baths or oral herbal medicine may precipitate hemolysis in affected children. Therefore, comprehensive education is essential at discharge for children at risk of or diagnosed with G6PD deficiency, emphasizing the avoidance of herbal remedies for jaundice treatment to mitigate the risk of G6PD-induced hemolysis.

Higher TSB levels at first discharge were identified as a significant risk factor for readmission in this study, consistent with findings by Chang et al [18]. Some neonates experience elevated bilirubin levels within 72 hours of phototherapy cessation, necessitating re-initiation of treatment [19,20]. However, Jodiery et al. found no significant differences in bilirubin levels between the time of phototherapy termination and 24-48 hours post-termination [21].

Our study has limitations, including its design as a single-center retrospective cohort study with a relatively small sample size. Additionally, the lack of comprehensive tracking of all discharged infants may mean that some individuals with rebound hyperbilirubinemia sought treatment at other healthcare facilities, potentially affecting the outcomes. Based on our results, we recommend prolonging hospitalization and implementing robust post-discharge monitoring for neonates with prematurity, ABO hemolysis, G6PD deficiency, or elevated total serum bilirubin levels at discharge.

## Supporting information

**S1 File. Data.**
(XLSX)

## Author contributions

**Data curation:** Ping Wang, Yanyan Song.

**Writing – original draft:** Yueju Cai, Xiaolan Li.

**Writing – review & editing:** Yueju Cai, Xiaolan Li.

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
