## [Decision Letter · Decision Letter 0]

2 Jan 2025

PONE-D-24-45603Analysis of readmission risk factors for neonatal hyperbilirubinemia

PLOS ONE

Dear Dr. cai,

Thank you for submitting your manuscript to PLOS ONE. After careful consideration, we feel that it has merit but does not fully meet PLOS ONE’s publication criteria as it currently stands. Therefore, we invite you to submit a revised version of the manuscript that addresses the points raised during the review process.

**ACADEMIC EDITOR: **

In addition to addressing the comments given by the reviewers, I would like to ask you address the following important issues: 

Abstract section:

Write abbreviations in full in their first use in the text. Eg. Total Serum Bilirubin (TSB)

Introduction section:

Which is the appropriate term, “nuclear jaundice Vs Kernicterus”.  Page- 1, paragraph -1, line -4.I appreciate the conciseness of the introduction section. However, it will be good if the authors mention findings of previous similar studies.

Participants and methods section:

Why gestational age of 35 weeks is taken as cut-off point for inclusion in the study? Why not 34 weeks or 37 weeks?Babies born at 35 weeks do not have the same bilirubin metabolism capacity to that of babies born at term. Have you controlled for the variable “gestational age” in the analysis when you are comparing for the effect of other variables between the two groups?The authors used the term “Retrospective” to describe their study design. However, the term “retrospective” doesn’t describe type of study design and it only tells us that the data used in the study is past time event. Therefore, it is better to use terms such as “retrospective cross-sectional study” or “retrospective cohort study” or “case-control study” design as deemed necessary.Provide explanation why you excluded babies with congenital malformation, hypoxia, infectious disease…. from the study while you included babies with G6PD deficiency and ABO hemolytic disease.In the introduction section, the authors have stated that the aim of the study was to determine “…risk factors for readmission…”. However, it appears that the authors implemented a “Retrospective cross-sectional” study design which is not suitable to determine temporal relationship between variables which is important to determine “risk factors”. Cross-sectional study allows to determine “association” between two variables and doesn’t allow determination of “risk factors”. Moreover, “risk” is calculated by using “incidence rate” which not assessed in the current study. Therefore, I recommend the authors to think about the appropriateness of the use of the term “…risk factor…”. Also, think about the title; which one is appropriate title for the data set you have used: “analysis of readmission risk factors” Vs “Predictors of readmission”?The authors mentioned that “Phototherapy and exchange transfusion are performed in accordance with established guidelines.” I advise the authors to cite these guidelines so that readers can refer them.Move the “ethical approval statement” to the end of the methods section.

Table 1:

Abbreviations used the table should be written in full or should be written in expanded form as foot noteThe variable “Resuscitation at Birth” has “zero value” in all the cells in the table. Therefore, there is no need to list this variable in the table.

Table 2:

The variables “Exchange transfusion” and “Rh hemolysis” have “zero value” in all the cells in the table. Therefore, there is no need to list these variables in the table.

Table 3:

Table-3 lacks clarity since it is not clear how some of the variables are categorized. For example, how is the variable “Birth weight” is categorized and its Odds Ratio is calculated?The variable “Gestational age” is included in the regression model twice (as continuous variable and as a categorical variable). Such inclusion of a single variable in different form distorts the regression model and the authors either have to remove one form of the variable or provide plausible explanation for doing so.The variables “Gestational age” and “birth weight” appears to have collinearity. However, the authors included both variables in the model while collinear variables shouldn’t be included without checking that they are no collinear.What is the basis of comparing the variables “preterm birth <37 wks” vs “Term birth”? We know that preterm babies are at increased risk of hyperbilirubinemia due to immaturity of UDGT enzyme.

We look forward to receiving your revised manuscript.

Kind regards,

Atnafu Mekonnen Tekleab, M.D

Academic Editor

PLOS ONE

2. In the online submission form, you indicated that [The datasets used and/or analysed during the current study available from the corresponding author on reasonable request].

Reviewers' comments:

Reviewer's Responses to Questions

**Comments to the Author**

1. Is the manuscript technically sound, and do the data support the conclusions?

Reviewer #1: Yes

2. Has the statistical analysis been performed appropriately and rigorously? 

Reviewer #1: Yes

3. Have the authors made all data underlying the findings in their manuscript fully available?

Reviewer #1: Yes

4. Is the manuscript presented in an intelligible fashion and written in standard English?

Reviewer #1: Yes

5. Review Comments to the Author

Reviewer #1: It is important to pick such kind of topics which is not uncommon to see newborn readmitted for after receiving treatment for neonatal hyperbilirubinemia. The research also reflects what common seen in our daily clinical activities.

The tittle, objectives, methodology, data analysis and conclusion are clearly stated and acceptable. The only concern that I have in this paper is small sample size( the readmission group), that make difficult to generalized the finding and draw strong recommendation.

6. PLOS authors have the option to publish the peer review history of their article (what does this mean? ). If published, this will include your full peer review and any attached files.

**Do you want your identity to be public for this peer review?** For information about this choice, including consent withdrawal, please see our Privacy Policy .

Reviewer #1: No

---

## [Author Response · Author response to Decision Letter 1]

20 Feb 2025

In response to your feedback, we have carefully revised the manuscript and provided detailed responses to each of the reviewers' comments in the file of response to reviewers. We truly appreciate your efforts and thoughtful recommendations, and we hope the revisions meet your expectations.

---

## [Editor Report · Decision Letter 1]

25 Feb 2025

Predictive Factors for Readmission Due to Neonatal Hyperbilirubinemia: A Retrospective Case-Control Study

PONE-D-24-45603R1

Dear Dr. Cai,

We’re pleased to inform you that your manuscript has been judged scientifically suitable for publication and will be formally accepted for publication once it meets all outstanding technical requirements.

Kind regards,

Atnafu Mekonnen Tekleab, M.D, MPH

Academic Editor

PLOS ONE
---

## [Editor Report · Acceptance letter]

PONE-D-24-45603R1

PLOS ONE

Dear Dr. cai,

I'm pleased to inform you that your manuscript has been deemed suitable for publication in PLOS ONE. Congratulations! Your manuscript is now being handed over to our production team.

Kind regards,

on behalf of

Dr. Atnafu Mekonnen Tekleab

Academic Editor

PLOS ONE